# Using Importance–Performance Analysis (IPA) to Improve Golf Club Management: The Gap between Users and Managers' Perceptions

Virginia Serrano-Gómez [1], Oscar García-García [1,*] and Antonio Rial-Boubeta [2]

[1] Faculty of Education and Sports Sciences, Universidad de Vigo, 36005 Pontevedra, Spain; vserrano@uvigo.es
[2] Faculty of Psychology, Universidad de Santiago de Compostela, 15782 Santiago de Compostela, Spain; antonio.rial.boubeta@usc.es
[*] Correspondence: oscargarcia@uvigo.es

**Abstract:** This work is carried out in order to provide new evidence on the usefulness of the Importance–Performance Analysis (IPA) in the management of golf courses, identifying the main strengths, weaknesses, and discrepancies between managers and users. This information will make it possible to identify potentially mistaken beliefs of managers regarding the service and to design improvement strategies based on the results. The participants were 11 managers, with an average age of 35.4 years (σ 6.5), of whom 72.7% were men and 27.3% were women, and 891 users, with an average age of 47.5 years (σ 12.3), of whom 81.7% were men and 18.3% were women. The instrument used was the Q-Golf scale, and among others, it was applied to the IPA. The results obtained show how the perception of managers regarding the service is worse than that of users. Additionally, the global assessment of the users is higher than the average assessment of each of the items of the club. This positive halo effect, which is not found in the case of managers, is very interesting from the point of view of the loyalty of golf users.

**Keywords:** industrial golf; golfers; loyalty; assessment of the sports service; user's satisfaction; manager's perception; strategies management





## 1. Introduction

The literature reveals how users usually evaluate the performance of a service based on a limited number of items, of which the most important will affect, to a greater extent, the general assessment, providing an indirect measure of satisfaction [1].

In this process, there is an average tolerance zone between the level of service desired by customers and an adequate level. Thus, performance assessments, understood as being within the tolerance performance zone, entail an evaluation of overall service satisfaction. Levels below the tolerance zone will cause customer frustration, reduce loyalty, and result in dissatisfied customers. Conversely, levels above the tolerance zone will lead to customer satisfaction and increased loyalty [2].

The limits of the customer's tolerance zone can be modified during the provision of the service since they are dynamic [3]. For this reason, managers must carry out monitoring and control processes continuously to obtain periodic information on the status of the items. Consequently, it will be essential to use effective measurement tools that allow knowing not only the valuation of the service but also the importance given to each of the different attributes.

The Importance–Performance Analysis (IPA), originating from Martilla and James [4], is a very useful method to determine the factors to be studied as a matter of priority. The IPA grid, derived from the analysis of the dimensions of importance and performance, can identify areas that need improvement or with excess resources the dimensions of importance and performance, can identify areas that need improvement or with excess

resources. The position of each attribute in the quadrant will depend on their average scores in the importance and assessment variables, where "Importance (I)" would reflect the relative value placed on the service, and "Performance (P)" the perception of the performance of a service [5]. Each of the quadrants of the graph is associated with a recommendation regarding the actions to be taken with each attribute. The *Focus here* quadrant constitutes the weak points of the service and main areas for improvement, bringing together all those attributes that are important but are not well valued. *Keeping up the good work* is a strong point of the service, so it does not require corrective actions by the organization, concentrating on the important and highly valued attributes. *Low priority* is an attribute with low importance and low valuation, so it is not necessary to pay much attention since it does not decisively influence the customer's evaluation. *Possible waste of resources* gathers well-valued items, but they are hardly important to them; in this way, it is possible that too many resources are being dedicated to it [6].

In relation to the limitations of the IPA, it is considered for the improvement of the differences between the results obtained in the assessment and the importance, which provides more information [7]. Besides, it was also considered the representation of the IPA based on the information provided by the diagonal [8,9]. Diagonal models separate the IPA space into two triangular halves, better predicting the priorities expressed by the users. Thus, the results of the discrepancies that are located in the upper part of the diagonal may require corrective actions, while the elements located below the diagonal will obtain positive evaluations. Therefore, the subjects will be satisfied with the attributes located below the diagonal and more dissatisfied with those that are in the upper zone.

In this way, the use of the IPA has continued to grow in recent years, being used in studies in many different areas, such as health, to understand the needs of the relatives of critically ill patients [10], the restaurant industry, to know the perspective of the consumer [11], tourism, to evaluate the tourist's satisfaction in the observation of fauna [12] or the evaluation for sustainable tourism [13], the economic, in studies on farms [14], etc.

Likewise, the use of the IPA is also growing in the sports sector for the assessment of fitness services [15–19], in sports centers according to public or private ownership [20], for tennis users [21] and baseball spectators [22], at hockey camps [23], sporting events [24,25], or managers and students of sports management [26], could be some examples.

More specifically, the IPA has also been used in the field of golf as a management tool [27] and for evaluation of golf customer satisfaction [28], for the study of the quality of service of a virtual reality center [29], to identify the most important aspects in young consumers of golf clothing [30], to evaluate the most important aspects of the condition of the golf course turf among consumers and turf managers [31], etc. However, there are no studies on the perception of golf managers and users that compare the importance and valuation of both with respect to the elements of the service.

In this line, the importance of golf is increasing; the number of golfers in the world has increased in recent years to 66.6 million, and of these, 10.6 million golfers play regularly on the European continent [32].

In Spain, the number of members also continues to grow, and currently (2023) reaches 293,560 federation licenses, translating into the fourth consecutive annual increase [33]. The R&A and Sports Marketing Surveys [32] consider golf the engine of the recovery of tourism and the Spanish economy and point out the following keys to the industry of this sport in Spain: it attracts around 1.2 million foreign tourists per year; it produces 11,183 million euros (generated directly or induced by foreign tourists); it benefits sectors other than golf (7 out of every 8 euros produced, 88%, benefit other sectors); the golf tourist spends more and stays longer, deseasonalizing tourism since its peak seasons are in spring and autumn. It also generates 121,393 direct and indirect quality jobs (95% are permanent and 94% full-time).

For all of the above, this work is carried out with the aim of providing new evidence on the usefulness of IPA in golf course management, identifying the main strengths and weaknesses of the service, and comparing the discrepancies in the importance and perfor-

mance of the items that form the Qgolf scale, between managers and users. In an applied way, it will be useful to know the managers' perception of the service and contrast them with the users' opinions in order to identify possible erroneous beliefs about the service offered, correct them, and design improvement strategies based on the results.

## 2. Materials and Methods

### 2.1. Participants

The participants were 891 users and 11 managers of golf clubs from the northwest of Spain. The average age of the users was 47 years ($\bar{x}$ 47.6; σ 12.3), of whom 81.7% were men and 18.3% women. Golfers of all levels were included; the majority (55.4%), showing handicaps below 18.4, were members of a club (59.7%) and played weekly (87.5%). Likewise, the average age of the managers was 35 years ($\bar{x}$ 35.4; σ 6.5), of whom 72.7% were men and 27.3% women. Most of the managers have a university education (72.7%); however, only 36.4% have a specialization degree in golf. A total of 54.6% started playing golf at the age of 25, and 81.8% have a handicap below 16.3.

### 2.2. Procedure

The procedure chosen for data gathering in every case was a personal and structured interview through a questionnaire that included the validated QGolf scale [34] based on a 5-point Likert–type response format (1–5). Each interview took place in the facilities of the participant clubs, with the authorization of the managers. Respondents were selected by convenience sampling. Each interview lasted approximately 15 min and was carried out by an external staff (researchers and assistant researchers), not related in any way to the clubs, properly trained to do that, and with expertise in that kind of study.

The research was approved by the board of directors of the golf courses, as well as by the local ethics committee. Participants answered the questionnaire on a voluntary basis, ensuring the anonymity and confidentiality of the data. Informed consent was obtained individually from all study participants. In addition, the principles of the Declaration of Helsinki on Human Research-64th World Medical Assembly 2013 were respected.

### 2.3. Measures

The instrument for collecting the information was a questionnaire with the QGolf scale of perceived quality for golf clubs by Serrano-Gómez et al. [34]. It is made up of 15 items and 3 dimensions obtained from a Confirmatory Factor Analysis. Its internal consistency is high overall ($\alpha$Global= 0.91) for each dimension ($\alpha$Staff = 0.88; $\alpha$Facilities = 0.80, and $\alpha$Course/Play zone = 0.79, and a high ability to explain user satisfaction ($R2$ = 0.72).

From the items that make up the scale, the adaptation of the Importance–Performance Analysis (IPA) of Abalo et al. [6] was used, where the discrepancies between the valuation and the importance given by users and managers were represented.

### 2.4. Data Analysis

Data analysis included descriptive statistics, means, standard deviations, and discrepancies. In addition, to compare the importance and performance scores of users and managers, the Student *t*-test was used. The non-compliance with the assumption of normality of the data was confirmed by the K-S Lilliefors test, which, together with the presence of some outliers and the large difference in the sizes of the two subsamples to compare, advised a complementary use of a non-parametric test (Mann–Whitney). Likewise, an Importance–Performance chart was performed, with the discrepancy scores from managers and users together, analyzing their situation respecting the diagonal and different quadrant components.

IBM SPSS Statistics 25 package was used to calculate statistical analyses.

### 3. Results

In the first place, Tables 1 and 2 are shown for descriptive purposes only. They are not intended to test significant differences between the scores of importance and performance. As shown in Table 1, the elements to which users give more importance are the State of the facilities (4.55), the Design and round of the course (4.46), followed by the Organization of tournaments (4.45) and the Professionalism of the Master caddie (4.44). Three additional elements cause importance greater than 4.40: Changing rooms (4.42); Management professionalism (4.41); and Course safety (4.41).

**Table 1.** Importance, Performance, and Discrepancy for users (n = 891).

| Dimension/Item | | Item | Average Performance (Standard Deviations) | Average Importance (Standard Deviations) | Discrepancy |
|---|---|---|---|---|---|
| Staff and Management | I1 | Management professionalism | 3.67 (1.20) | 4.41 (0.81) | −0.74 |
| | I2 | The professionalism of the reception staff | 3.99 (0.97) | 4.31 (0.80) | −0.32 |
| | I3 | The professionalism of the Master caddie | 4.09 (0.97) | 4.44 (0.72) | −0.41 |
| | I4 | Organization and management of resources | 3.58 (1.05) | 4.40 (0.73) | −0.82 |
| | I5 | Communication management | 3.61 (1.10) | 4.19 (0.81) | −0.58 |
| | I6 | Complaints and suggestions management | 3.44 (1.17) | 4.12 (0.87) | −0.68 |
| | I7 | Organization of tournaments in club | 4.02 (0.95) | 4.45 (0.73) | −0.43 |
| Facilities | I8 | State of club facilities | 3.97 (0.89) | 4.55 (0.64) | −0.58 |
| | I9 | State of furnishings and materials | 3.76 (1.01) | 4.36 (0.74) | −0.6 |
| | I10 | Clubhouse/Social hall | 3.68 (1.20) | 4.28 (0.79) | −0.6 |
| | I11 | Changing rooms | 3.59 (1.20) | 4.42 (0.75) | −0.83 |
| Course | I12 | Control of play and rules compliance | 3.58 (1.13) | 4.37 (0.83) | −0.79 |
| | I13 | Design and round of the course | 4.11 (0.86) | 4.46 (0.68) | −0.35 |
| | I14 | Course safety | 3.80 (0.97) | 4.41 (0.79) | −0.61 |
| | I15 | Practice area | 3.89 (0.96) | 4.36 (0.74) | −0.47 |
| Average Importance of the different elements | | | | 4.37 (0.49) | |
| Average Performance of the different elements | | | | 3.79 (0.70) | |
| Global average value of the service | | | | 4.05 (0.64) | |

In the case of managers (Table 2), the most important elements (with averages above 4.40) are as follows: Organization of the tournament (4.45); the Professionalism of the Master caddie (4.44); the Professionalism of the reception staff (4.45); and Course safety (4.45). The average importance that both give to the set of service elements does not present significant differences (t = 0.62; *p* = 0.63). Nor are significant differences detected in any specific element of the service. Only the difference regarding the importance they give to the Clubhouse/Social hall (Table 3) borders on statistical significance (t = 1.906; *p* = 0.057), being users who give more importance to this element (4.28 vs. 3.82).

**Table 2.** Importance, Performance, and Discrepancy for Managers (n = 11).

| Dimension/Item | | Item | Average Performance (Standard Deviations) | Average Importance (Standard Deviations) | Discrepancy |
|---|---|---|---|---|---|
| Staff and Management | I1 | Management professionalism | 4.00 (1.00) | 4.27 (0.79) | −0.27 |
| | I2 | The professionalism of the reception staff | 4.18 (0.75) | 4.45 (0.69) | −0.27 |
| | I3 | The professionalism of the Master caddie | 3.73 (0.65) | 4.45 (0.82) | −0.72 |
| | I4 | Organization and management of resources | 3.73 (0.79) | 4.36 (0.67) | −0.63 |
| | I5 | Communication management | 3.00 (0.63) | 4.27 (0.79) | −1.27 |
| | I6 | Complaints and suggestions management | 3.45 (1.04) | 4.18 (0.75) | −0.73 |
| | I7 | Organization of tournaments in club | 3.91 (0.83) | 4.45 (0.82) | −0.54 |
| Facilities | I8 | State of club facilities | 3.45 (0.93) | 4.36 (0.67) | −0.91 |
| | I9 | State of furnishings and materials | 3.55 (1.13) | 4.18 (0.75) | −0.63 |
| | I10 | Clubhouse/Social hall | 3.45 (1.04) | 3.82 (1.17) | −0.37 |
| | I11 | Changing rooms | 3.27 (1.49) | 4.36 (0.67) | −1.09 |
| Course | I12 | Control of play and rules compliance | 3.73 (0.79) | 4.18 (1.25) | −0.45 |
| | I13 | Design and round of the course | 4.55 (0.52) | 4.18 (0.87) | 0.37 |
| | I14 | Course safety | 4.18 (0.60) | 4.45 (0.93) | −0.27 |
| | I15 | Practice area | 3.64 (1.03) | 4.09 (0.83) | −0.45 |
| Average Importance of the different elements | | | | 4.27 (0.46) | |
| Average Peformance of the different elements | | | | 3.72 (0.50) | |
| Global average value of the service | | | | 3.73 (0.65) | |

**Table 3.** Comparative Importance of Users vs. Managers.

| Dimension/Item | | Item | Discrepancy (Users—Managers) | t Value [1] | Z Value [2] |
|---|---|---|---|---|---|
| Staff and Management | I1 | Management professionalism | 0.14 | 0.58 | −0.77 |
| | I2 | The professionalism of the reception staff | −0.15 | −0.60 | −0.50 |
| | I3 | The professionalism of the Master caddie | −0.10 | −0.04 | −0.21 |
| | I4 | Organization and management of resources | 0.03 | 0.15 | −0.33 |
| | I5 | Communication management | −0.08 | −0.33 | −0.27 |
| | I6 | Complaints and suggestions management | −0.06 | −0.22 | −0.06 |
| | I7 | Organization of tournaments in club | 0.01 | 0.01 | −0.16 |
| Facilities | I8 | State of club facilities | 0.18 | 0.94 | −1.10 |
| | I9 | State of furnishings and materials | 0.18 | 0.80 | −0.93 |
| | I10 | Clubhouse/Social hall | 0.46 | 1.91 | −1.41 |
| | I11 | Changing rooms | 0.06 | 0.26 | −0.51 |
| Course | I12 | Control of play and rules compliance | 0.18 | 0.72 | −0.18 |
| | I13 | Design and round of the course | 0.28 | 1.35 | −1.20 |
| | I14 | Course safety | −0.04 | −0.18 | −0.53 |
| | I15 | Practice area | 0.27 | 1.21 | −1.22 |

[1] Student t value, [2] Z Mann–Whitney value; $p < 0.05$.

Regarding the assessment (Table 1) of the different elements of the service, in the case of users, only three elements exceed the score "4": the Design and round of the course (4.11), the Professionalism of the Master caddie (4.09), and the Organization of tournaments (4.02), precisely three elements that are especially important for users.

In the case of managers, again, there are only three elements that exceed the "4": Design and round of the course (4.55); the Professionalism of the reception staff (4.18); and the Course safety (4.18). The average assessment that both give to the set of elements of the service does not show significant differences (t = 0.34; $p$ = 0.73). Nor are significant differences detected in practically any specific element of the service. Only the difference regarding the assessment of Communication management (Table 4) is significant (t = 3.12; $p < 0.05$), being users who make a better assessment (3.61 vs. 3). The State of the facilities also presents a relatively important difference, although in this case, it is not significant (t = 1.91; $p$ = 0.056). Once again, the evaluation of the users is better than that of the managers themselves (3.97 vs. 3.45).

**Table 4.** Comparative Performance of Users vs. Managers.

| Dimension/Item | | Item | Discrepancy (Users—Managers) | t Value [1] | Z Value [2] |
|---|---|---|---|---|---|
| Staff and Management | I1 | Management professionalism | −0.33 | −0.92 | −0.46 |
| | I2 | The professionalism of the reception staff | −0.19 | −0.66 | −0.49 |
| | I3 | The professionalism of the Master caddie | 0.36 | 1.22 | −1.83 |
| | I4 | Organization and management of resources | −0.14 | −0.45 | −0.17 |
| | I5 | Communication management | 0.61 | 3.12 * | −2.27 * |
| | I6 | Complaints and suggestions management | −0.01 | −0.04 | −0.19 |
| | I7 | Organization of tournaments in club | 0.11 | 0.37 | −0.66 |
| Facilities | I8 | State of club facilities | 0.52 | 1.91 | −1.83 |
| | I9 | State of furnishings and materials | 0.21 | 0.70 | −0.47 |
| | I10 | Clubhouse/Social hall | 0.23 | 0.62 | −0.93 |
| | I11 | Changing rooms | 0.32 | 0.88 | −0.97 |
| Course | I12 | Control of play and rules compliance | −0.15 | −0.43 | −0.25 |
| | I13 | Design and round of the course | −0.44 | −1.67 | −1.63 |
| | I14 | Course safety | −0.38 | −1.30 | −1.22 |
| | I15 | Practice area | 0.26 | 0.83 | −0.91 |

[1] Student t value, [2] Z Mann–Whitney value; * $p < 0.05$.

Finally, in the case of users, it is worth noting (Table 1) the difference found between the global assessment they make of the Club (4.05) and the assessment resulting from averaging the 15 elements evaluated (3.79), finding a positive "halo effect", which is not found (Table 2) in the case of managers (3.73 and 3.72, respectively).

Regarding the Discrepancies between Importance and Performance (Figure 1), it is important to highlight that in both cases, the major part of the discrepancy scores is negative, which shows that for both targets, the performance would be below the demands of the subjects. Only in the case of managers does one element show a positive discrepancy, the Design and round of the course (Discrepancy = 0.37).

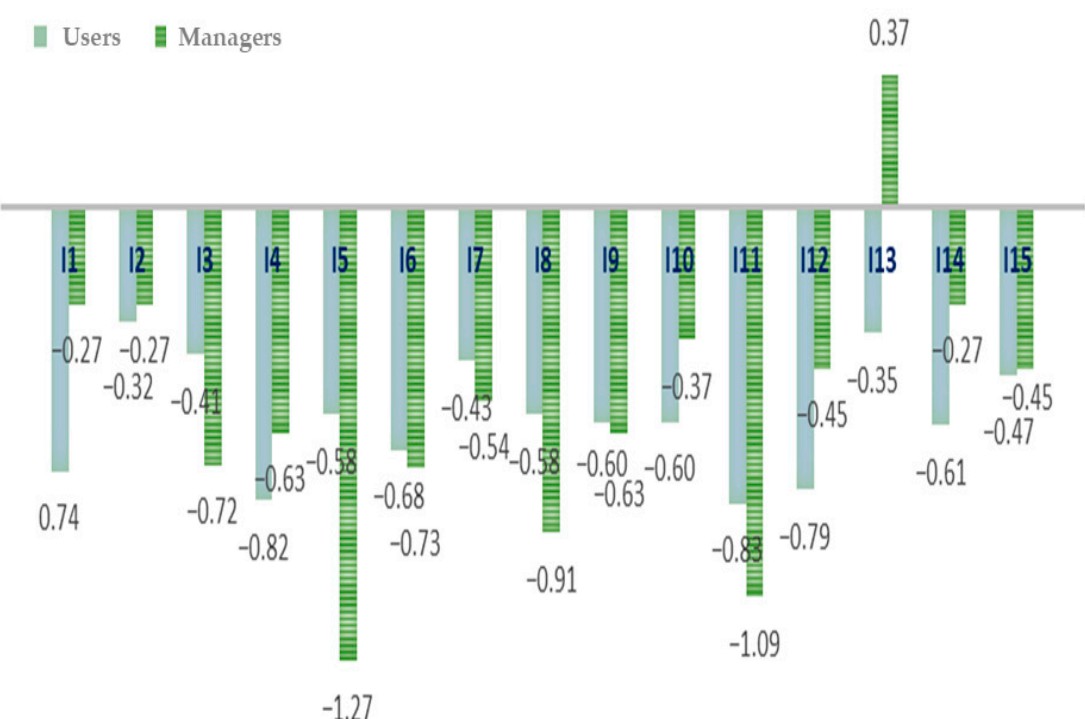

**Figure 1.** Comparative Discrepancies in Importance–Performance of Users vs. Managers.

Beyond this difference, Figure 1 shows some issues of interest. It is possible to identify three elements in which, according to the perception of the users, the performance would not be as unsatisfactory or deficient as for the managers themselves, finding much greater negative discrepancies in the case of the latter: I5- Communication Management (−0.58 vs. −1.27); I8-State of club facilities (−0.58 vs. −0.91); and I11-Changing rooms (−0.83 vs. −1.09).

Conversely, three elements are identified, in which, according to the managers' perception, the performance would not be as deficient as for the users themselves: I1-Management Professionalism (−0.74 vs. −0.27); I14- Course safety (−0.61 vs. −0.27); or I12-Control of play and rules compliance (−0.79 vs. −0.45). Therefore, these three items (I1, I14, I12) that have turned out to be more unsatisfactory for users than managers believed should be specially considered.

All these scores can be taken to the corresponding IPA Graph (Figure 2), being able to verify that some elements change position (and even quadrant) if we start from the perceptions of managers or users.

Thus, for example, element I13—Design and round of the course (the second more important for users)—would be in the "Keep up the good work" quadrant, considering the users' perceptions, while in the managers' opinion, it would be a "Possible waste of resources"; this difference would imply a gap to consider. Additionally, element I14—Course safety—is found in "Keep up the good work" for both managers and users; however, there is a greater distance from the diagonal in the case of users, and therefore, more dissatisfaction, so they really should not neglect this aspect.

On the other hand, it should be noted that the elements are located in the improvement area, so the priority for management in their case would be much higher. One of the cases corresponds to item I11—Changing rooms—both the manager and the users agree on the need for improvement, even the managers showing greater dissatisfaction, given that their score is further from the diagonal. Another item in the area of improvement corresponds to I12—Control of play and rules compliance. Here it is important to note that users are dissatisfied with this item, while for managers, it was a low priority, so they clearly need to react and take action. In the same way, the element I4—Organization and management of

the resources—is an element to improve (and that is how users consider it as managers); however, the dissatisfaction is even greater on the part of the users. Lastly, element 1— Management Professionalism—is also considered an area for improvement, but curiously, according to the perception of the managers, this element is in the "Possible waste of resources" quadrant. Based on these results, managers must carry out self-criticism and consider how they can improve.

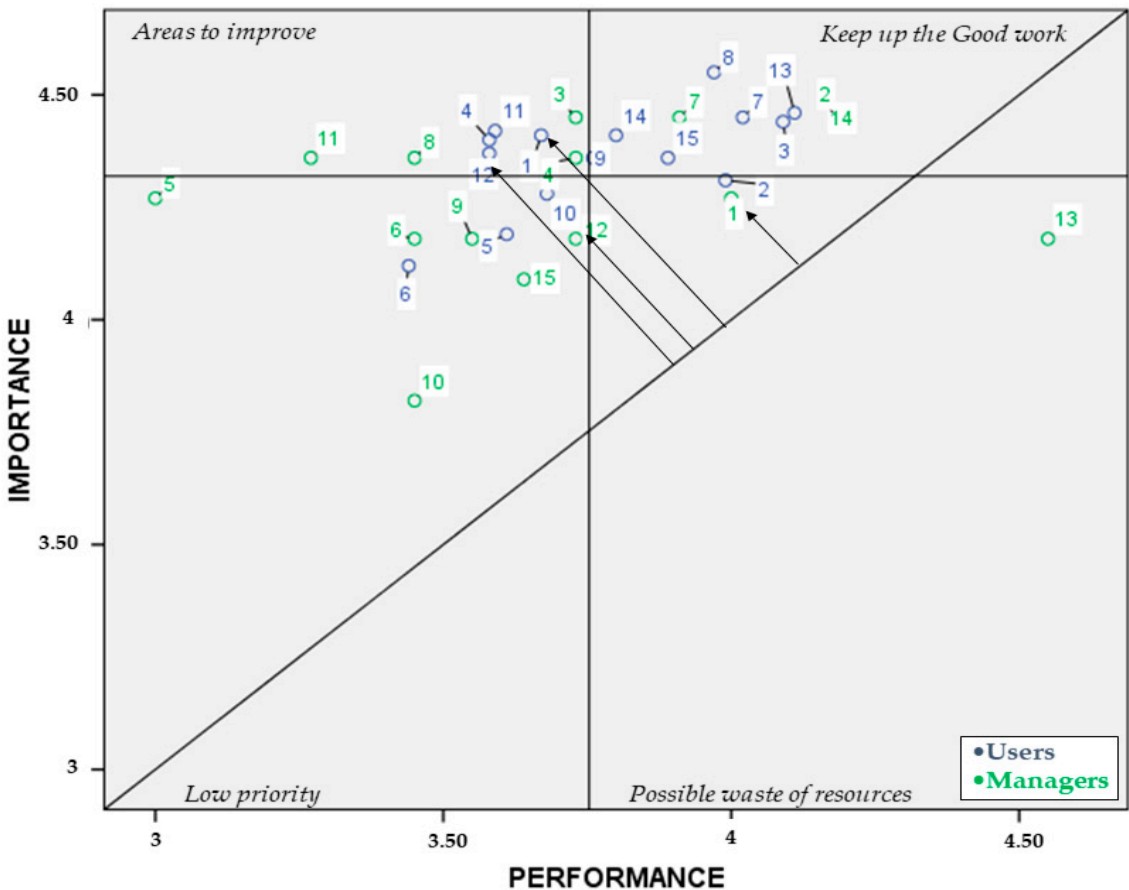

**Figure 2.** Comparative IPA graph Users vs. Managers.

## 4. Discussion

This work is carried out with the aim of providing new evidence on the usefulness of the IPA in golf course management, identifying the main strengths and weaknesses, and comparing the discrepancies in the importance and performance of the items that form the Qgolf scale between managers and users. In an applied way, it will be useful to know the managers' perception of the service and contrast them with the users' opinions in order to identify possible erroneous beliefs about the service offered, correct them, and design improvement strategies based on the results.

The use of IPA in the sports sector has increased notably in recent years [15,18,19,22,25], including in golf [28,29]. However, although there are studies applying this type of analysis in golf user participants [27], there are no references in the literature that use the IPA on the perception of golf managers.

In this case, if the number of managers sampled from the study might seem limited a priori, it could really be considered representative of the sector since it was made up of 50% of the total managers of golf facilities in Galicia. In this sense, different works have shown the value that IPA analysis provides in different settings, even when used with small samples, especially if the participants are experts or managers [9,35].

Regarding the managers interviewed, the majority were men (72.7%), a male reality that also coincides with most users (81%), and managers of the majority of sports organizations in general, which continue to be led by men [36]. More specifically, even the percentage of male golf course managers may be higher, 91.67% [37] or 92.9% [38], as seen in similar studies.

The results obtained indicate the Organization of tournaments, the Professionalism of the Master caddie, the Professionalism of the reception staff, and Course safety as the most important elements for golf course managers. However, for the users, the most important thing is the State of the facilities, the Design and round of the course, followed by the Organization of tournaments, and the Professionalism of the Master caddie aspects clearly related to sports practice. It is observed that both agree on the importance of organizing tournaments and on the professionalism of the Master caddie. However, the manager does not consider the State of the facilities or the Design and round of the course as important as the users. These conclusions coincide with other works in the sports sector, for example, in fitness centers [5,17], where the personnel, equipment, and state of the facilities are usually one of the most important elements for users.

Likewise, managers should not neglect the Clubhouse/Social hall, which is important for golf users and, on the contrary, is less relevant for managers. In this line, Quesada and Gómez-López [39] point out among the main motivations of users of sports centers the desire to have fun and the promotion of social relationships. This reality on the part of most social users should not be neglected. Precisely, Serrano-Gómez et al. [40] point to having fun and exercising as the main reasons for playing golf, the lack of time and work reasons being the main barrier to more practice. These results are similar to those obtained by Shim et al. [41], who consider such motivating factors significant for golf adherence. In this line, four types of golf users could be distinguished and grouped, in turn, into two segments, Competitive (Regulars and Leisure and Business) and Social (Matures and Familiars) [40]. The so-called Social seem to be the most numerous group (70% of users); therefore, if this is usual, it would be appropriate to review the strategies used and also contemplate other more playful and family actions aimed at greater participation of this group, beyond the ownership of the golf courses (social, commercial, mixed, etc.).

With respect to the most valued items, both managers and users agree on the design and round of the course. These results reinforce what was suggested by other authors [42,43]: the dimensions that best explain user preferences are generally the conditions and facilities of the golf course. For this reason, the work of the Greenkeeper is gaining more and more relevance, not being limited only to garden care, but his function has been evolving toward the responsibility of the design/construction of the golf course itself [44]. On this point, Pradas-García and García-Tascón [45] agree on the need for good communication with the direction/management of the course, reporting the concerns and needs of the users in order to respond to them. Bearing this in mind, we agree with the authors, pointing out that better course maintenance will help attract a greater number of users, and in view of the results, better control of the rules and regulations (Master caddie) will help ensure adherence.

The foregoing also agrees with the other two items most valued by users, the Professionalism of the Master caddie and the Organization of tournaments, so in this section, those managers seem to be doing a good job. However, it should be remembered that the needs and characteristics of those interested are multiple, and the gain of one group could also translate into the loss of another [46]. This simply reinforces the need to identify, analyze and classify those interested in order to design appropriate strategies in each case.

Significantly, it is observed the item communication management; in this case, managers give considerable importance to this element but perform poorly. This shows how managers are more critical than users. Likewise, the Global assessment of the users is greater than the average assessment of each of the elements of the club. This positive "halo effect", which is not found in the case of managers, is very interesting from a loyalty point

of view. In this way, the entity will be temporarily favored by reducing the effect [47], in this case, of the most negative items.

Despite this halo effect and apparently positive ratings, the evidence of the IPA analysis shows that the performance would be below the demands of the subjects. This situation is also found in similar studies on the use of the IPA [6] in the sports context [16,17,21], in which managers need to pay special interest to items in the superior half of the triangle, and in particular, those furthest away from the discrepancy line, such as Organization and management of the resources, Changing rooms, Control of play and rules compliance, and Management professionalism. The elements directly related to management, considered by users as items to be improved; however, they have not been for managers themselves, who place themselves in the "Possible waste of resources" quadrant.

## 5. Conclusions

Based on the results, managers should carry out self-criticism and consider new strategies to improve these aspects of management, which seem to generate some dissatisfaction among users. These aspects must be reviewed in a particular way, and each manager must carry out periodic monitoring and control of the service in the managed golf course through tools such as the IPA. The feedback received will allow the necessary action measures to be carried out based on objective data and not impressions that could be misleading.

Customer satisfaction is a requirement for service quality, and in this case, understanding the connection between the importance of the items and their valuation through the experiences of internal and external customers seems essential to expand the competitive advantage of the entity and increase the probability of survival of golf courses in the long term.

Future studies with an even larger sample of managers and varied geographically will allow us to continue this line of work on monitoring and control tools and techniques in the field of sports management. Given their dynamic nature, scales such as QGolf must continue to be tested, including, among other things, qualitative studies, and incorporate into it, as appropriate, other elements that current events demand. Finally, the increasing use of techniques, such as the IPA, and its variants, should continue to be studied in an applied way due to its unquestionable value in the performance evaluation process.

**Author Contributions:** V.S.-G., O.G.-G. and A.R.-B. contributed to this work in equal measure. This manuscript was drafted, revised, and commented on by all authors. All authors have read and agreed to the published version of the manuscript.

**Funding:** This research received no external funding.

**Institutional Review Board Statement:** The principles of the Declaration of Helsinki on Human Research-64th World Medical Assembly 2013 were respected.

**Informed Consent Statement:** Informed consent was obtained from all subjects involved in the study.

**Data Availability Statement:** Data is unavailable due to privacy or ethical restrictions.

**Conflicts of Interest:** The authors have no conflict of interest.

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
