# Peer review of "Using Importance–Performance Analysis (IPA) to Improve Golf Club Management: The Gap between Users and Managers’ Perceptions"

_sustainability, doi:10.3390/su15097189_

Round 1
Reviewer 1 Report
Dear Authors/Editors,
Thank you for the opportunity to review “Using Importance-Performance Analysis (IPA) to improve golf club management: the gap between users and managers’ perceptions”. I believe the topics fit within the scope of “Sustainability". The issue is innovative, timely, and interesting, but the paper has some shortcomings that should be addressed if the paper is to be published. The authorship of the article presents a very high number of authors, which may constitute a weakness for publication.
This paper intended to provide new evidence on the usefulness of the Importance Performance Analysis (IPA) in the management of golf courses, identify the main strengths and weaknesses, and compare the results obtained by managers and users.
The paper presents a good literature review, considering the main keywords (importance-performance analysis (IPA); industrial golf; loyalty; assessment of the sports service; user´s satisfaction; manager´s perception; strategies management) and indicators selected. The Methodology and Data Analysis are correctly structured and presented. However:
- The abstract should present a better understanding/relevance of the topic to be investigated.
- The final section in the introduction should be improved with a view to better exposing the structure of the article and specifying the adopted methodology.
- Sub-chapters should never end with a table, or a figure as happens in section 3. They should end with a complimentary analysis.
- The number of references used to support the literature review is not reflected in the presented discussion.
- Regarding Discussion, this section is well structured. However, the analysis should be deepened in order to deepen Theoretical and Practical Implications and propose strategies and actions to the organizations in order to ensure possible solutions’ discussion, considering the main research results.
Author Response
Thank you for the opportunity to review “Using Importance-Performance Analysis (IPA) to improve golf club management: the gap between users and managers’ perceptions”. I believe the topics fit within the scope of “Sustainability". The issue is innovative, timely, and interesting, but the paper has some shortcomings that should be addressed if the paper is to be published.
R. Dear reviewer, thank you very much for your comments.
The authorship of the article presents a very high number of authors, which may constitute a weakness for publication.
R. We have read carefully your comment and perhaps we have not understood you correctly, we think that 3 authors are not a very high number for the publication.
This paper intended to provide new evidence on the usefulness of the Importance Performance Analysis (IPA) in the management of golf courses, identify the main strengths and weaknesses, and compare the results obtained by managers and users.
The paper presents a good literature review, considering the main keywords (importance-performance analysis (IPA); industrial golf; loyalty; assessment of the sports service; user´s satisfaction; manager´s perception; strategies management) and indicators selected. The Methodology and Data Analysis are correctly structured and presented.
R. Thank you very much for your comment.
The abstract should present a better understanding/relevance of the topic to be investigated.
R. Thank you for this suggestion. We have introduced some changes in the abstract, both in the objective and in the participants that improve the understanding of the study.
The final section in the introduction should be improved with a view to better exposing the structure of the article and specifying the adopted methodology.
R. Following your suggestion, we have modified and specified the purpose of the article at the end of the introduction for a better understanding.
Sub-chapters should never end with a table, or a figure as happens in section 3. They should end with a complimentary analysis.
R. Following your suggestion, we have modified the place of the text by analyzing figure 2, at the end of section 3.
The number of references used to support the literature review is not reflected in the presented discussion.
R. A. Following your advice, we have included some more references from the literature review in the discussion.
Regarding Discussion, this section is well structured. However, the analysis should be deepened in order to deepen Theoretical and Practical Implications and propose strategies and actions to the organizations in order to ensure possible solutions’ discussion, considering the main research results.
R. Thank you very much for your comment. In this sense, we have included a new paragraph in the conclusion that better explains the action that managers should take based on the results.
Reviewer 2 Report
Dear authors,
Thank you very much for this interesting paper. Although, I believe that several considerations should be taken into account
Abstract: the number of participants must be included (mean age and standard deviaton)
Keywords: try to include different words because the title has the same in some cases (ie IPA)
Introduction: generally within the text when we talk about the authors it is better to put AND (Martilla and James). You write 293,560 federation licenses but from 2023, 2022…?
Procedure: you use a hoc questionnaire. I consider it important that this questionnaire has gone through a validation process (at least by expert judges and understanding for the participants). Was any pre-testing of the instrument performed before it was passed?
Measures: Serrano-Gómez et al. [3.4] please fix [3,4]. In addition, you could include in the questionnaire the scale (1 to 6, 1 to 4…) and the initial premise to pass the questionnaire
Data analysis: Why were parametric (t-student) and non-parametric (Z Mann Whitney?) procedures mixed? I would like to know if any type of exclusion criteria was used to eliminate atypical cases and how the verification was carried out if the data were parametric or non-parametric
In the results, a descriptive analysis is made indicating the mean and the difference between both values ​​without statistical significance (Table 1-2). But in table 3, you can see the value of T and Z without statistical significance in any of the data (p <.05) although this does appear in the legend, I think it should be removed. Only one significant piece of information is seen in “communication management”. I believe that this data should be reflected in the results as it is the only significant one.
Line 220 and 218, 212, 209, the format is not correct (the size is reduced). It also happens on the next page.
Ensure that the figure (Areas to improve, keep up the Good work) are in Palatino Lotyle font
The discussion is adequate, but I think that if "communication management" was the only real data that was significant, it is important to highlight this aspect.
Include in all references the DOI of all those that are available.
Author Response
Dear authors, Thank you very much for this interesting paper. Although, I believe that several considerations should be taken into account.
Abstract: the number of participants must be included (mean age and standard deviaton)
R. Dear Reviewer. Thank you very much for your comments. Regarding this point, we included the number of participants in the abstract and improved the text referring to the mean age of the participants and the standard deviation.
Keywords: try to include different words because the title has the same in some cases (ie IPA)
R. Following your suggestion, we have removed the keyword “IPA” and have included “golfers”
Introduction: generally within the text when we talk about the authors it is better to put AND (Martilla and James).
R. We changed them. Thank you.
You write 293,560 federation licenses but from 2023, 2022…?
R. We have included the year 2023 in that paragraph.
Procedure: you use a hoc questionnaire. I consider it important that this questionnaire has gone through a validation process (at least by expert judges and understanding for the participants). Was any pre-testing of the instrument performed before it was passed? Measures: Serrano-Gómez et al. [3.4] please fix [3,4]. In addition, you could include in the questionnaire the scale (1 to 6, 1 to 4…) and the initial premise to pass the questionnaire
R. Thank you for this suggestion. This manuscript is the continuation of a line of research, where the reliability and validity of the instrument has already been evaluated (citation 34), with good results as included in the measures section: It is made up of 15 items and 3 dimensions obtained from a Confirmatory Factor Analysis. Its internal consistency is high both overall (αGlobal= .91), and for each dimension (αStaff = .88; αFacilities = .80; and αCourse/Play zone = .79, and a high ability to explain user satisfaction (R2=0.72).
However, we are very grateful for your comment and to make this section clearer, we added in the text that the instrument used was validated and the Likert-type point scale (1-5) used as you recommend: "The procedure chosen for data collection in all cases was a personal, structured interview, through a questionnaire including the validated QGolf scale [34], based on a 5-point Likert-type response format (1-5)".
In addition, we have corrected the typo [34]. Thank you.
Data analysis: Why were parametric (t-student) and non-parametric (Z Mann Whitney?) procedures mixed? I would like to know if any type of exclusion criteria was used to eliminate atypical cases and how the verification was carried out if the data were parametric or non-parametric
R. Regarding to Data Analysis, the wording of this section has been slightly modified, with appropriate justification for the complementary use of parametric and nonparametric tests, as well as the treatment of outliers.
In the results, a descriptive analysis is made indicating the mean and the difference between both values ​​without statistical significance (Table 1-2). But in table 3, you can see the value of T and Z without statistical significance in any of the data (p <.05) although this does appear in the legend, I think it should be removed.
R. Regarding to Results, tables 1 and 2 are for descriptive purposes only; they are not intended to test the possible significant differences between the levels of importance and valuation. This was not the aim of the work. Therefore, no legend on the p-values is included, nor the word "Comparative" in the title of the tables and no statistical tests were run. A comment in this sense has been added in the text (page 4). In the case of tables 3 and 4, the objective was the statistical comparison between the two targets (users and managers). Although in table 3 there is no statistically significant difference, we chose to keep the legend with p-value in both tables.
Only one significant piece of information is seen in “communication management”. I believe that this data should be reflected in the results as it is the only significant one.
R. This information is included in the results, and we have also included a paragraph in the discussion on communication management.
Line 220 and 218, 212, 209, the format is not correct (the size is reduced). It also happens on the next page.
R. The reduced format has been changed.
Ensure that the figure (Areas to improve, keep up the Good work) are in Palatino Lotyle font
R. Figure 2 has been corrected, and all the texts are in Palatino Linotype font.
The discussion is adequate, but I think that if "communication management" was the only real data that was significant, it is important to highlight this aspect.
R. A specific comment highlighting the difference in “communication management” has been included in the Discussion.
Include in all references the DOI of all those that are available.
R. In response to your suggestion. 8 new DOIs have been added to the references.
Reviewer 3 Report
I think that the article raises a very interesting topic and is quite well written. It may be a bit of a wonder why such a number of users and 11 managers took part in the study, what was the reason for this? Well, as if it was explained in the research part of the article. In general, it should be said that the article is short, because without bibliography and tables/graphs it is only about 6-7 pages, which is not a complaint, I just point it out. I also believe that it is worth explaining clearly in the introduction what is the purpose of the article, so that it is understandable to a wide audience. I think that's missing at the moment. In conclusion, I think that the article is interesting, correctly written and deserves publication.
Author Response
I think that the article raises a very interesting topic and is quite well written.
R. Thank you very much for your comment.
It may be a bit of a wonder why such a number of users and 11 managers took part in the study, what was the reason for this?
R. In Galicia (Northwestern Spain) there are 22 golf courses, of which 11 managers voluntarily attended our request to participate in this research.
Well, as if it was explained in the research part of the article. In general, it should be said that the article is short, because without bibliography and tables/graphs it is only about 6-7 pages, which is not a complaint, I just point it out. I also believe that it is worth explaining clearly in the introduction what is the purpose of the article, so that it is understandable to a wide audience. I think that's missing at the moment.
R. Regarding the length of the article, we appreciate your comment, this manuscript is the continuation of previous works. The objective on this occasion is more specific, including as a novelty the perception of golf course managers and their relationship with the opinion of users through the IPA.
In this sense, following your suggestion, we have modified and specified the purpose of the article at the end of the introduction for a better understanding.
In conclusion, I think that the article is interesting, correctly written and deserves publication.
R. Thank you very much for your comment.
Reviewer 4 Report
The objective of the article is to study the measurement tools (specifically the API) that allow to evaluate the differences in satisfaction of a service (here golf) between customers and managers. The methodology used is quantitative and qualitative with a high number of respondents per questionnaire and per interview.
On reading the paper, some criticisms must be pointed out:
Concerning the tables and figures, the authors should add a legend in order to facilitate reading and understanding.
The results show the important elements that determine the satisfaction of users and managers. Very practical reasons are provided by the authors (facilities, installations, reception, etc.). On the other hand, other reasons, perhaps less obvious, are not mentioned by the authors. For example, the choice of a golf course according to the user population is an explanatory element. It also corresponds to a search for differentiation and/or identification of users with social classes. These sociological reasons are absent from the article (lines 295 and 296 for example). This is a pity.
The article remains overall very descriptive. It would need more in-depth reflection on the expectations of the new sporting public. Sport is not only a leisure activity, it is also a social activity that the text does not study.
Author Response
The objective of the article is to study the measurement tools (specifically the API) that allow to evaluate the differences in satisfaction of a service (here golf) between customers and managers. The methodology used is quantitative and qualitative with a high number of respondents per questionnaire and per interview. On reading the paper, some criticisms must be pointed out:
R. Dear reviewer, thank you very much for your comments. We have read carefully your comments and suggestions. Based on them we have made changes that we believe that improves the manuscript.
Concerning the tables and figures, the authors should add a legend in order to facilitate reading and understanding.
R. We appreciate your comment, however the meaning of each element is already included in the introduction: "Importance (I)" would reflect the relative value attributed to the service. "Performance (P)" the perception of the performance of a service. Discrepancies between the scores obtained in the assessment and the importance.
The results show the important elements that determine the satisfaction of users and managers. Very practical reasons are provided by the authors (facilities, installations, reception, etc.). On the other hand, other reasons, perhaps less obvious, are not mentioned by the authors. For example, the choice of a golf course according to the user population is an explanatory element. It also corresponds to a search for differentiation and/or identification of users with social classes. These sociological reasons are absent from the article (lines 295 and 296 for example). This is a pity.
R. Thank you for your comments. This work is the continuation of previous works where this objective is already addressed. On this occasion, the purpose is different, including as a novelty the perception of golf course managers and their relationship with the opinion of users through the IPA. Likewise, we understand the importance of your comment and therefore this topic is already discussed between lines 279 and 288.
The article remains overall very descriptive. It would need more in-depth reflection on the expectations of the new sporting public. Sport is not only a leisure activity, it is also a social activity that the text does not study.
R. As we explained in the previous commentary. This work is the continuation of previous works where this objective is already addressed. On this occasion, the purpose is different, including as a novelty the perception of golf course managers and their relationship with the opinion of users through the IPA.
Round 2
Reviewer 2 Report
Thank you for your effort,
Please, check the right use one English and some mistakes like blond lines (271-285).
Reviewer 4 Report
The authors explained their methodology and answered each point correctly. The document is better in its current form.